# High-Risk Genetic Multiple Myeloma: From Molecular Classification to Innovative Treatment with Monoclonal Antibodies and T-Cell Redirecting Therapies

**DOI:** 10.3390/cells14110776

**Published:** 2025-05-25

**Authors:** Danilo De Novellis, Pasqualina Scala, Valentina Giudice, Carmine Selleri

**Affiliations:** 1Department of Medicine and Surgery “Scuola Medica Salernitana”, University of Salerno, 84081 Baronissi, Italyvgiudice@unisa.it (V.G.); 2Hematology and Transplant Center, University Hospital “San Giovanni di Dio e Ruggi d’Aragona”, 84131 Salerno, Italy

**Keywords:** multiple myeloma, molecular mechanism, genomic stratification, high-risk genetic MM, anti-CD38 agents, CAR-T, BiTEs, proteasome inhibitors, CD38, BCMA, MRD

## Abstract

High-risk genetic multiple myeloma (HRMM) remains a major therapeutic challenge, as patients harboring adverse genetic abnormalities, such as del(17p), *TP53* mutations, and biallelic del(1p32), continue to experience poor outcomes despite recent therapeutic advancements. This review explores the evolving definition and molecular features of HRMM, focusing on recent updates in risk stratification and treatment strategies. The new genetic classification proposed at the 2025 EMMA meeting offers improved prognostic accuracy and supports more effective, risk-adapted treatment planning. In transplant-eligible patients, intensified induction regimens, tandem autologous stem cell transplantation, and dual-agent maintenance have shown improved outcomes, particularly when sustained minimal residual disease negativity is achieved. Conversely, in the relapsed or refractory setting, novel agents have demonstrated encouraging activity, although their specific efficacy in HRMM is under investigation. Moreover, treatment paradigms are shifting toward earlier integration of immunotherapy, and therapeutic strategies are individualized based on refined molecular risk profiles and clone dynamics. Therefore, a correct definition of HRMM could help in significantly improving both clinical and therapeutic management of a subgroup of patients with an extremely aggressive disease.

## 1. Introduction

MM is a clonal hematological malignancy characterized by the uncontrolled proliferation of malignant plasma cells in the bone marrow, associated with overproduction of monoclonal immunoglobulins (M-proteins), resulting in end-stage organ damage [1]. Over the last decade, the OS of MM patients has significantly improved, largely due to the advent of innovative therapeutic strategies, including monoclonal antibodies and T-cell redirecting therapies [2]. Despite these advances and the use of novel treatments, approximately 15-20% of patients continue to experience poor outcomes, with a median OS shorter than three years. These patients typically exhibit aggressive disease biology driven by specific high-risk genetic abnormalities, and they represent a clinically distinct subset, defined as high-risk multiple myeloma (HRMM), that could overlap with a recently introduced entity, the functional high-risk multiple myeloma (FHRMM), which includes patients with early relapse or suboptimal response to therapies. [3]. To date, the presence of these adverse prognostic high-risk genetic features remains a major unmet clinical challenge, even in the era of novel agents. While several reviews have discussed therapeutic options in MM, few have comprehensively addressed the specific challenges posed by HRMM, such as risk classification systems and specific therapeutic strategies. Therefore, in this review, we aim at (i) outlining the historical evolution and updated definitions of HRMM; (ii) summarizing current therapeutic landscapes, including the role of monoclonal antibodies and T-cell redirecting therapies; and (iii) highlighting future directions, such as molecularly guided MRD-adapted treatment approaches.

## 2. High-Risk Genetic Abnormalities

An overview of high-risk cytogenetic abnormalities in MM is summarized in Table 1. The presence of these abnormalities at diagnosis remains one of the principal barriers to long-term efficacy with current anti-myeloma therapies.

The gold standard for detecting these genetic alterations in approximately 90% of patients is FISH analysis performed on CD38^+^-enriched bone marrow plasma cells [4]. Using this technique, several cytogenetic and molecular abnormalities associated with HRMM have been identified, including translocations involving the *IGH* locus (e.g., t(4;14), t(14;16), and t(14;20)), del(17p), gain or amplification of 1q, and deletion of 1p [5]. Overall, hyperdiploidy and chromosomal abnormalities involving the *IGH* locus (14q32 region) are detected in ~45–50% of MM cases. In detail, t(11;14) is found in ~20% of patients and is considered a standard-risk abnormality; conversely, t(4;14) (10–15% of cases), t(14;16) (~5%), and t(14;20) (~5%) are less frequent and are associated with high-risk genetic disease due to their correlation with poorer outcomes [6,7].

Chromosome 1 abnormalities are the most frequent cytogenetic alterations in newly diagnosed MM, occurring in 30–40% of cases, and are considered important prognostic markers. In particular, 1q gain and 1q amplification, defined as the presence of one or ≥2 additional copies of the 1q region [8], have a proportional adverse prognostic impact that increases with the number of additional 1q21 copies, as 1q21 amplification is associated with a worse outcome compared to simple gain [9,10]. Deletions of the 1p32 region are rare, as they are detected in ~10% of newly diagnosed MM cases; however, they represent a strong and independent adverse prognostic factor, especially in cases of biallelic deletions [11,12]. Loss of the short arm of chromosome 17—del(17p)—is the worst adverse prognostic factor in MM, particularly when co-occurring with *TP53* mutations, a condition known as “double-hit *TP53* multiple myeloma” [13]. The prevalence of del(17p) ranges from 8% to 15%, depending on the threshold used to define the proportion of clonal plasma cells, as a cut-off of ≥55% del(17p)-positive plasma cells is the only value significantly linked to worse prognosis [14,15]. The functional loss of TP53 results in profound genomic instability and is strongly associated with pan-resistance to anti-myeloma agents, as MM cells become largely insensitive to apoptosis [16].

## 3. Evolution of High-Risk Genetic Classification

In 2015, the standard ISS was revised to incorporate adverse cytogenetic abnormalities, including t(4;14), t(14;16), t(14;20), del(17p), and *TP53* mutations, as well as elevated serum LDH values, leading to the creation of the R-ISS [17]. Subsequently, in 2016, the IMWG further refined the definition of HRMM by including additional cytogenetic abnormalities, such as 1q gain/amplification and 1p32 deletions, which were not yet formally integrated into the R-ISS, leading to a discrepancy between these two staging systems [3]. This gap highlights the urgent need to revise current risk stratification models by incorporating the full spectrum of genetic abnormalities with validated prognostic relevance. In this context, a novel risk stratification system has been proposed and validated by Hervé Avet-Loiseau at the 30th EMMA meeting held in Vienna in January 2025 [18]. This updated model demonstrated superior prognostic accuracy compared to the IMWG system, identifying del(17p) with a cut-off >20% of positive cells, *TP53* mutations, and biallelic del(1p32) as independent negative prognostic markers. Additionally, specific combinations were identified as high-risk, including chromosome 14 translocations co-occurring with 1q gain or monoallelic del(1p32) and 1q21 gain combined with monoallelic del(1p32) (Table 2) [18]. If definitively validated, the new genetic risk model will allow a deeper understanding of HRMM, and therapies can be tailored based on specific patient characteristics.

## 4. Molecular Mechanisms in HRMM

In HRMM, the presence of specific molecular alterations confers aggressive disease behavior, drug resistance, and clonal evolution [19], which can be pharmacologically targeted. Del(17p), with or without *TP53* mutations, results in the loss of p53 tumor suppressor function, promoting genomic instability, impaired DNA repair, and apoptosis resistance, thereby contributing to unresponsiveness to conventional therapies [20]. Gain or amplification of chromosome 1q21, another hallmark of HRMM, involves several oncogenes, which accelerate cell cycle progression, drug resistance, and adverse outcomes [21]. *IGH* translocations lead to altered chromatin structure and epigenetic regulation, DNA damage tolerance, resistance to proteasome inhibitors, cell adhesion, migration, and angiogenesis (Figure 1) [12,22]. These molecular events not only drive disease progression but also induce an immunosuppressive microenvironment, impacting the efficacy of monoclonal antibodies and T-cell redirecting agents. A deeper understanding of these mechanisms is essential for the development of targeted therapeutic strategies tailored to the genetic and molecular landscape of HRMM [23,24].

In detail, there are different pathogenetic events occurring at disease initiation and at progression (Figure 2) [25,26].

### 4.1. IGH Translocations

Translocations involving the *IGH* loci place various oncogenes next to the strong enhancer region of these immunoglobulin regions, which are highly active in mature B cells; thus, translocated oncogenes result in increased expression, ultimately resulting in cell cycle dysregulation and proliferation and reduced DNA repair ability [27,28]. *IGH* translocations and hyperdiploidy usually are present at disease initiation and lead to cell cycle dysregulation by affecting CDK4 and CDK6 complexes that favor the dissociation transcription factor E2F by RB phosphorylation [29,30,31]. Consequently, E2F concurs to transcription of genes involved in the G1/S checkpoint step and to upregulation of FGFR3 and MMSET (also known as NSD2) [32,33]. This latter, a histone methyltransferase, highly influences the methylation status of cells by increasing H3K36me2 and reducing H3K27me3 [34]. In neoplastic plasma cells, MMSET upregulation leads to altered methylation, cell adhesion, increased proliferation and survival, and genomic instability [35,36]. Moreover, MMSET can promote non-homologous end-joining at deprotected telomeres, altering the DNA repair process [37]. Other *IGH* translocations, including t(14;16) and t(14;20), are associated with overexpression of the oncogenes MAF and MAFB, integrins such as integrin β7, and apolipoprotein B mRNA editing enzyme catalytic subunit-induced mutation signature [38,39]. All these alterations are linked to tumor invasion, metastasis, higher ability to resist starvation and stress, and increased genomic and chromosomal instability [37].

### 4.2. Methylation

HDAC6 inhibition results in high pro-apoptotic effects, likely because of a concomitant modulation of protein degradation. HDAC6 binds to polyubiquitinated proteins and facilitates the removal of protein aggregates by regulating aggresome formation and their autophagic degradation through HSF1 and HSP90 activation [40]. This abnormal methylation status in neoplastic plasma cells is also caused by increased expression of the histone methyltransferase EZH2 [30]. Moreover, mutations in DNA methylation modifiers, such as in IDH1, can also occur, especially during disease progression, and can contribute to global gene expression alteration [30].

### 4.3. MYC Translocations

Secondary events at disease progression are *MYC* translocations, likely promoted by genes with super-enhancers active in late B cell stages [41]. Copy number changes usually occur at 8q24 and are associated with MYC translocation in 30% of cases [42,43].

### 4.4. Chromosome 1 Abnormalities

Amplification of 1q is also common and starts from the formation of dicentric chromosomes, leading to multiple breakage–fusion–bridge cycles with consequent gene at 1q21 [44]. This chromosomal region comprises several genes, including CKS1B, MCL1 encoding for a BCL-2 family member, ANP32E, and ILF2 encoding for a protein required for RNA splicing of genes for DNA damage repair proteins [45]. These alterations are probably induced by hypoxia and aberrant expression of KDM4A, as also demonstrated by a significant association between *RAS* mutations, loss of p53 function, and upregulation of HIF1α and lactate dehydrogenase A [46,47]. Conversely, loss of 1p leads to deletion of CDKN2C, FAF1, FAM46C, RPL5, and ecotropic viral integration site 5 [48]. In particular, CDKN2A, together with CDKN2B, CDKN2C, and CDKN2D, regulates cell proliferation by inhibiting the activity of cyclin D–CDK complexes [49]. Their loss of function can derive from 1p deletion, homozygous inactivation of RB1, and/or DNA methylation-mediated silencing, and results in increased cell proliferation alongside RAS pathway upregulation [30]. Therefore, chromosome 1 abnormalities result in cell cycle dysregulation, proliferation, anti-apoptotic signaling activation, and cell growth [37].

## 5. First-Line Treatments for Transplant-Eligible Patients

In Europe, MM treatment strategy is tailored according to the patient’s fitness and eligibility for ASCT, as outlined in the 2021 EHA-ESMO guidelines (Table 3) [50]. The primary goal of frontline treatment is to achieve an early, profound, and sustained response, ideally defined as high sensitivity (at least 10^−5^) MRD negativity. Consequently, treatment strategies involving intensified induction regimens, ASCT, consolidation, and maintenance are designed to maximize the depth and duration of response. Sustained MRD negativity is critical, as transient MRD negativity does not correlate with improved long-term outcomes [51]. Before the advent of anti-CD38-based induction, 4-6 cycles of VRD represented the most effective frontline option for HRMM patients [3,52,53]. Afterwards, the addition of daratumumab, an anti-CD38 monoclonal antibody, to dara-VTD demonstrated excellent responses, resulting in FDA and EMA approval in 2019 and 2020, respectively [54]. HRMM patients treated with dara-VTD showed favorable CR and negative MRD rates compared to VTD only [55], as well as daratumumab plus VRD (dara-VRD), which further increased negative MRD rates [56]. This consistent benefit of dara-VRD in HRMM has been further confirmed in the phase III Perseus trial [57,58] and in the IFM phase II 2018-04 trial. In the latter, four cycles of quadruplet induction and consolidation therapy with dara-KRd, followed by tandem ASCT, and two years of maintenance with daratumumab plus lenalidomide, resulted in promising outcomes for HRMM [59]. Isatuximab, another anti-CD38 monoclonal antibody, in combination with KRD (isa-KRd), has also demonstrated high rates of MRD negativity post-consolidation in both high- and ultra–high-risk (≥ 2 high-risk cytogenetic abnormalities) MM patients in the randomized phase III ISKIA trial [60], and in the GMMG-CONCEPT study both in transplant-eligible and ineligible HRMM patients [61].

Efficacy of tandem ASCT in HRMM has been demonstrated in several studies (Table 4) [62,63,64,65,66], and current international guidelines recommend double ASCT in this setting [50,67]. However, these studies have been conducted in the pre-daratumumab era, when MRD monitoring was not routinely employed to guide treatment strategies. Therefore, it remains uncertain whether tandem ASCT continues to provide a significant advantage in the context of modern induction regimens incorporating anti-CD38 monoclonal antibodies. To address this question, the ongoing phase 3 Minimal Residual Disease Adapted Strategy (MIDAS) trial (NCT04934475) is evaluating the role of single versus tandem ASCT in the era of anti-CD38–based therapies. After a uniform induction consisting of six cycles of Isa-KRd, patients are stratified into four cohorts based on MRD status to determine the optimal transplantation strategy.

Consolidation therapy is defined as 2-4 additional cycles of treatment using the same agents employed during induction followed by ASCT, and its role in MM management remains debated. For example, VRD-based regimens have been proposed as consolidation because some studies, like EMN02/HO95, display improved outcomes, while others, such as the StaMINA trial, have not confirmed this benefit [64,66,68]. However, current clinical guidelines recommend the use of consolidation therapy in selected cases, such as patients with persistent MRD positivity after ASCT or those with HRMM. Given the limited and heterogeneous evidence regarding consolidation, maintenance therapy has emerged as a critical component in post-transplant management, particularly in HRMM, where sustained disease control is essential for improving long-term outcomes. Maintenance treatment is less intensive than induction, typically administered orally until MM progression or intolerance, with the primary aim of delaying relapse [69,70]. Lenalidomide alone has shown clear benefits in PFS and OS in the general MM population [71], although its efficacy in HRMM appears limited in prolonging PFS. Therefore, lenalidomide in combination with bortezomib, with or without dexamethasone, could offer added benefit in this subgroup of patients [7,72,73]. In the CASSIOPEIA trial, single-agent daratumumab has been shown to prolong PFS; however, no specific subgroup analysis was reported for HRMM [74]. Conversely, the randomized phase III TOURMALINE-MM3 study demonstrated that ixazomib as maintenance after ASCT could improve outcomes in the HRMM group [75]. In the ongoing PERSEUS study (NCT03710603), maintenance with lenalidomide and daratumumab is being tested in both standard and high-genetic risk patients [57,76]. Despite these promising results, clinical trials lack patients’ stratification by high-risk genetic features or only include limited analyses on HRMM disease, partly due to the underrepresentation of high-risk patients in trial populations. Additionally, the absence of MRD-guided treatment adaptation in several older pivotal studies limits their relevance in the current precision medicine landscape.

## 6. First-Line Treatment for Transplant-Ineligible Patients

In ASCT-ineligible patients, the therapeutic goal shifts away from achieving a deep and sustained response by balancing treatment efficacy with tolerability and preserving quality of life. However, evidence-guided treatment decisions in this setting are limited, particularly for frail patients with high-risk genetic abnormalities, who are often underrepresented in clinical trials (Table 5).

Bortezomib in combination with IMiDs has been associated with hematological responses in HRMM patients [77]. In the phase III VISTA trial, the VMP regimen demonstrated comparable survival rates to those observed in standard-risk patients [78]. In contrast, continuous or fixed-duration RD therapy is ineffective in these patients [79]. However, adding bortezomib to RD as a VRD triplet can improve PFS [80], whereas the addition of elotuzumab did not show the same benefit [81]. Moreover, integration of anti-CD38 monoclonal antibodies into frontline regimens has yielded mixed results in HRMM. Indeed, dara-RD has documented superiority over RD in HRMM, although the adverse prognostic impact of high-risk genetics is not fully mitigated, as PFS remains shorter than that observed in the overall study population [82,83,84]. Similarly, the addition of daratumumab to the VMP regimen did not improve outcomes compared to VMP in HRMM patients [85]. Interpretation of these results is limited by the small proportion of high-risk patients in the intention-to-treat populations, reducing the statistical power of subgroup analyses. To address this limitation, a meta-analysis of phase III studies has been conducted, showing that the addition of daratumumab could prolong PFS in HRMM patients [86].

Alternative proteasome inhibitor–based combinations have also been explored in this setting. Carfilzomib plus cyclophosphamide-dexamethasone (KCD) may mitigate genetically related poor prognosis [87], and the quadruplet isa-KRD has been associated with encouraging MRD negativity rates at the end of consolidation [61]. However, in the phase III IMROZ trial, isatuximab plus VRD (isa-VRD) has not shown improved outcomes in patients with high-risk cytogenetic features and older age [88], as well as daratumumab plus VRD in the CEPHEUS study [89].

## 7. High-Risk Genetic Relapsed/Refractory MM

The prevalence of high-risk genetic features increases at MM progression due to clonal evolution after prior therapies [90]. While triplet combinations generally outperform doublets in both standard- and high-risk patients, their efficacy in HRMM remains suboptimal. Most regimens do not fully mitigate the adverse prognostic impact of these genetic abnormalities, and PFS consistently remains shorter compared to standard-risk patients (Table 6) [50].

According to ESMO guidelines, treatment selection is primarily based on lenalidomide sensitivity or refractoriness [50]. In lenalidomide-sensitive patients, triplet regimens, such as dara-RD, KRD, elotuzumab-RD, or ixazomib-RD, have shown clinical efficacy [91,92,93,94,95], although HRMM patients display inferior PFS and response durability compared to standard-risk cohorts. In lenalidomide-refractory HRMM patients, preferred regimens are combinations of anti-CD38 monoclonal antibodies with PD (dara-PD) or KD (isa-KD) [96,97,98,99]. In both phase III CASTOR and ICARIA trials, dara-VD and isa-PD have been associated with longer PFS in HRMM patients compared to control arms, although the strength of this benefit is lower than that observed in standard-risk populations [24]. Emerging agents may provide incremental benefits in HRMM. The anti-BCMA antibody-drug conjugate belantamab-mafodotin in combination with VD could be even superior to dara-VD in this subgroup of patients [100]. Similarly, the nuclear export inhibitor, selinexor, when combined with VD (SVD), has also demonstrated superiority over VD in HRMM, as shown in the phase III BOSTON trial [101].

Despite some clinical efficacy, these regimens are not curative, as current treatment options fail to achieve deep and durable responses comparable to those observed in standard-risk MM. Therefore, unmet needs persist across the treatment continuum for HRMM patients, and a rational treatment sequencing strategy would ideally prioritize the use of the most effective agents as earlier therapy lines, before clonal evolution would reduce drug responsiveness. Tailored approaches based on molecular profiles, coupled with MRD-adapted therapy escalation or de-escalation, could further enhance outcomes in this high-risk population.

## 8. T-Cell Redirecting Therapy for Relapsed/Refractory MM

Innovative T-cell redirecting therapies targeting surface antigens on malignant plasma cells include the family of (BiTEs), and CAR-T cells [102]. Currently, two anti-BCMA BiTEs, teclistamab and elranatamab, and one anti-GPRC5D BiTE, talquetamab, are approved for treatment of relapsed/refractory MM. Data on their efficacy in HRMM patients remains limited, as only a small proportion of them are included in clinical trials (Table 7) [102].

For example, HRMM patients treated with teclistamab or elranatamab show slightly worse outcomes compared to the overall population, as described in the phase I-II MajesTEC-1 trial and in the phase II MagnetisMM-3 study, respectively [103,104], while talquetamab induces similar ORR between high- and standard-risk genetic populations, as documented in the MonumenTAL-1 study [105]. A combination of teclistamab or elranatamab with daratumumab could be more effective; however, updated data on HRMM subgroups are not available yet [106,107,108]. Of particular interest is the combination of teclistamab and talquetamab in the RedirecTT-1 study, enrolling 63 patients with triple-refractory MM. Although results for the HRMM subgroup are not yet available, they are highly expected due to the innovative nature of this dual BiTE approach targeting distinct antigens [109]. Moreover, several trials are currently ongoing to assess the efficacy of BiTEs combined with other anti-MM agents, such as NCT05243797, NCT05083169, NCT05090566, NCT04649359, NCT05317416, NCT04798586, and NCT05228470.

CAR-T cells, autologous genetically engineered T lymphocytes, are modified to express a chimeric antigen receptor that recognizes the BCMA antigen, mainly on neoplastic plasma cells. In MM, two anti-BCMA CAR-T cell products are currently approved: idecabtagene vicleucel (ide-cel) and ciltacabtagene autoleucel (cilta-cel) [102]. Ide-cel, the first FDA-approved anti-MM CAR-T product, has shown efficacy also in HRMM, although with lower ORR and shorter PFS compared to standard-risk patients, as shown in the phase 2 KarMMa study [74,110]. In HRMM patients with cilta-cel after 3 or more prior lines of therapy, ORR is also impressive; however, duration of response and PFS are shorter compared to standard risk, as described in the CARTITUDE-1 trial [111,112]. Moreover, cilta-cel with PVd or DPd has comparable efficacy in high and standard genetic risk MM patients, as observed in the phase 3 CARTITUDE-4 study [113]. Several clinical trials are currently ongoing to evaluate the efficacy of CAR-T cell products in earlier lines of therapy, including in HRMM, such as in NCT04923893, NCT05257083, NCT05393804, and NCT06045806 trials.

Despite these advancements and the undeniable efficacy of T-cell redirecting therapies, several real-world barriers reduce their wide use in HRMM patients, also as earlier treatment lines. Indeed, toxicity management remains complex, as cytokine release syndrome and neurotoxicity require specialized and experienced centers for quick identification and treatment. Additionally, access to CAR-T cells and BiTEs is limited by manufacturing logistics, center capacity, and costs, which currently negatively impact their early use in routine clinical practice [114]. Moreover, the lack of long-term follow-up data in HRMM does not yet permit assessment of the cost-efficacy of these therapies, thus adding evidence to support their early use in this high-risk population [102].

## 9. Conclusions

In recent years, significant advancements have been made in MM treatment, with the introduction of several novel therapeutic agents for newly diagnosed patients, regardless of ASCT eligibility, as well as for those with relapsed/refractory disease. At the same time, increasing emphasis has been placed on a more precise risk stratification system by incorporating not only clinical and laboratory biomarkers but also molecular and cytogenetic alterations [20]. Indeed, HRMM has emerged as a distinct biological and clinical entity, characterized by aggressive behavior and poor prognosis, often driven by complex genomic and transcriptomic signatures. Despite important therapeutic and diagnostic innovations, prognosis for these subjects remains poor, likely due to different pathogenetic mechanisms underlying this type of disease. For this reason, a paradigm shift is essential for improving clinical management of HRMM, as future strategies should focus on refining risk stratification models and tailoring therapies based on specific patients’ molecular features. For example, the risk-adapted model proposed by Avet-Loiseau offers a more accurate classification of HRMM and should be prospectively validated for clinical implementation. Indeed, this model could enable early identification of ultra–high-risk patients and help tailor treatment intensity accordingly. In addition, the prognostic impact of specific lesions, such as biallelic TP53 inactivation or 1p32 deletions, must be clearly delineated in clinical trials to guide treatment escalation. Therefore, it is essential to improve current risk stratification systems to not only include cytogenetic abnormalities and *TP53* mutational status but also to consider clonal hematopoiesis and other pathogenetic variants, as described in myeloid and lymphoid malignancies. Moreover, routine integration of MRD detection into treatment algorithms represents a critical future direction, as MRD negativity is a strong predictor of long-term outcomes, particularly in HRMM. In this subset of MM patients, durable responses are less common, and MRD could serve as a surrogate to modulate therapy, allowing intensification for non-responders or de-escalation to reduce toxicity in deep responders. However, prospective trials are urgently needed to validate MRD-guided treatment decisions, especially in high-risk settings.

While intensified therapeutic approaches are now accepted standards, they alone are not sufficient. The integration of immunotherapy, particularly T-cell redirecting agents, such as BiTEs and CAR-T cells, earlier into frontline treatment, in a genetically risk-adapted and guided approach, would be a desirable strategy to overcome traditional resistance mechanisms. These agents, ideally combined with checkpoint inhibitors or targeted molecules, could provide synergistic effects. However, barriers related to toxicity, accessibility, cost, and logistical complexity remain substantial, particularly when considering frontline or early relapse application. Ultimately, addressing the needs of HRMM will require a multi-omics-driven approach coupled with prospective clinical trials specifically designed for this subgroup, rather than relying on post hoc analyses from broader studies. Only through this tailored strategy can we significantly modify the natural history of HRMM and offer a real opportunity for long-term disease control in this group of patients.

## Figures and Tables

**Figure 1 cells-14-00776-f001:**
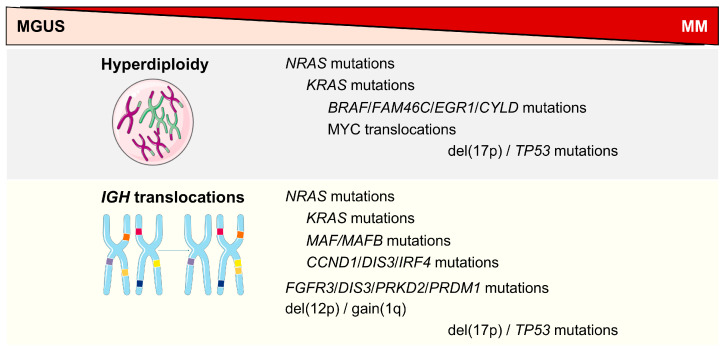
Genomic events at disease initiation and progression from MGUS to MM. Hyperdiploidy could be a first-hit event that facilitates the addition of secondary events, such as mutations in *NRAS*, *KRAS*, and *BRAF*. Similarly, translocations involving the immunoglobulin heavy chains (*IGH*) gene could be a different primary event that more frequently is associated with mutations in *MAF*/*MAFB*, *FGFR3*, or *CCND1* genes, as well as with gains of 1q. *TP53* mutations and/or del(17p) are usually late pathogenic events, occurring in both cases.

**Figure 2 cells-14-00776-f002:**
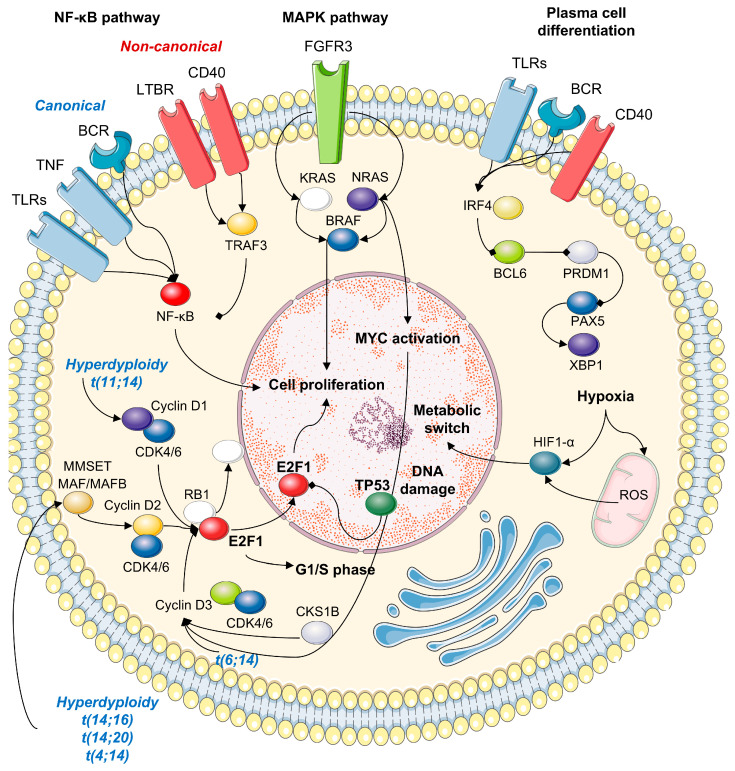
Pathways altered in MM. Principal pathways involved in high-risk disease, including NF-κB, MAPK, cell cycle, hypoxia, and DNA-damage repair pathways. In detail, the absence of TP53 results in the lack of cell cycle control through E2F1 and Cyclin D3, and cells continue to the G1/S phase even though they carry DNA breaks, thus promoting genomic instability, impaired DNA repair, and apoptosis resistance. The t(4;14), t(14;16), and t(14;20) translocations lead to MMSET and MAF/MAFB overexpression, ultimately leading to increased E2F1 activation and translocation to the nucleus, where it promotes the transcription of genes involved in cell proliferation. Similarly, hypodiploidy and t(11;14) promote cell cycle progression and proliferation through Cyclin D1/CDK4/9 modulation. Upregulation and/or activating mutations in FGFR3, KRAS, NRAS, and/or BRAF, and MYC translocations induce cell proliferation by direct gene transcription or by indirectly modulating cyclin activities, as well as TRAF3 and NF-κB. Moreover, altered BCR signaling is related to impaired apoptosis through modulation of IRF4, BCL-6, and PAX5. Finally, hypoxia can also influence RNA splicing of genes for DNA damage repair proteins via upregulation of HIF1α and lactate dehydrogenase A. Figure made using https://smart.servier.com/.

**Table 1 cells-14-00776-t001:** High-risk genetic abnormalities in multiple myeloma.

Abnormality	Frequency	Gene/Pathway	Prognostic Significance	Clinical Impact
All 14q32 (*IGH*) t(4;14) t(14;16) t(14;20)	45–50% 10% to 15% <5% <5%	FGFR3/MMSET Upregulation MAF overexpression MAFB overexpression	Poor Uncertain; mainly poor Uncertain; mainly poor	Rapid progression. Double ASCT Double ASCT, especially with HR abn Double ASCT, especially with HR abn
1q21 gain 2–3 copies ≥4 copies	40% 20–30% 5–20%	CKS1B, MCL1, ADAR1 overexpression	Intermediate Poor	Aggressive with organ failure Double ASCT, especially with HR abn
1p32 deletion Monoallelic Biallelic	10%	FAF1/CDKN2C deficit	Poor Highly poor	Double ASCT, especially with HR abn Double ASCT + intensive maintenance
del(17p)/*TP53* mutation Single hit Double hit	8–15% Deletion Deletion + mutation	TP53	Poor Highly poor	Poor sensitivity to therapy Double ASCT + intensive maintenance

**Table 2 cells-14-00776-t002:** Risk stratification systems.

Stage	ISS	R-ISS
I	Sβ2M < 3.5 mg/L Serum albumin ≥ 3.5 g/dL	Sβ2M < 3.5 mg/L Serum albumin ≥ 3.5 g/dL Standard-risk CA by iFISH Normal LDH
II	Sβ2M > 3.5 mg/L and serum albumin < 3.5 g/dL OR 3.5 mg/L < Sβ2M > 5.5 mg/L	Not R-ISS stage I or III
III	Sβ2M ≥ 5.5 mg/L	Sβ2M ≥ 5.5 mg/L and either high-risk CA by FISH OR high LDH
**Genetic risk**		
	R-ISS Standard-risk CA	Deletion of chromosome 17, or 17p-, translocation of chromosomes 14 and 16, or t(14;16), and translocation of chromosomes 4 and 14, or t(4;14)
	EMMA High genetic risk CA	del(17p) > 20% *TP53* mutation Biallelic del1p32 1q gain and monoallelic del1p32 t(4;14) or t(14;16) or t(14;20) and either 1q gain or monoallelic del1p32

**Table 3 cells-14-00776-t003:** First-line treatments in ASCT-eligible MM patients.

Study	Phase	Regimens	N. HRMM Patients	Outcomes in HRMM	General Outcomes	*p*-Value
CASSIOPEIA (Completed)	III	Dara-VTD vs. dara-VTD	168	CR rate: 36.6% MRD^−^ rate: 59.8% No benefits compared to VTD	CR or better: 38.9% MRD^−^ rate: 63.7%	NS
GRIFFIN (Completed)	II	Dara-VRD vs. dara-VRD	30	sCR rate: 18.8% MRD^−^ rate: 37.5% No benefits compared to VRD	sCR rate: 42.4% MRD^−^ rate: 51%	NS
PERSEUS (Active, not recruiting)	III	Dara-VRD vs. dara-VRD	154	MRD^−^ rate: 68.4% Sustained MRD^−^ rate: 48.7%	MRD^−^ rate: 75.2% Sustained MRD^−^ rate: 69.3%	0.04
ISKIA (Active, not recruiting)	III	Isa-KRD	111 Ultra-HR: 51	MRD^−^ rate: 76-77%	MRD^−^ rate: 79% in standard risk	NS
GMMG-CONCEPT (Active, not recruiting)	II	Isa-KRD	125	MRD^−^ rate in ASCT-eligible patients: 67.7% MRD^−^ rate in ASCT-ineligible patients: 54.2%	−	−
IFM 2018-04 (Active, not recruiting)	II	Dara-KRD	50	30-month PFS: 80% 30-month OS: 91% ORR: 100% in patients completing 2nd ASCT	−	−

**Table 4 cells-14-00776-t004:** Outcomes after ASCT in MM patients.

Study	Phase	Regimens	N. HRMM Patients	Outcomes in HRMM	General Outcomes	*p*-Value
STAMINA (Completed)	III	ASCT plus lenalidomide maintenance (auto/len) vs. ASCT plus VRd consolidation plus lenalidomide maintenance (auto/VRd) vs. tandem ASCT plus lenalidomide maintenance (auto/auto)	223	6-year PFS: 43.6% and 26% for auto/auto and auto/len	6-year PFS auto/auto: 43.9% auto/VRd: 39.7% auto/len 40.9% 6-year OS auto/auto: 73.1% auto/VRd: 74.9% auto/len 76.4%	0.03
EMN02/HO95 (Completed)	III	VCD followed by VMP or single/double ASCT	225	75-month OS 54% with ASCT 30% with VMP Median PFS double vs. single ASCT: 46 vs. 27.6 months	−	−
MIDAS (Active, not recruiting)	III	Isa-KRD × 6 plus in standard risk (A) Isa-KRD for 6 plus lenalidomide maintenance for 3 years (B) ASCT plus isa-KRD for 2 plus lenalidomide maintenance for 3 years (C) In high-risk, ASCT plus isa-KRD for 2 plus isa-Iber for 3 years (D) In high-risk, tandem ASCT plus isa-iber for 3 years			Ongoing	−

**Table 5 cells-14-00776-t005:** First-line treatments in ASCT-ineligible MM patients.

Study	Phase	Regimens	N. HRMM Patients	Outcomes in HRMM	General Outcomes	*p*-Value
SWOG S0777 (Active,not recruiting)	III	VRD vs RD	104	Median PFS: 38 months vs. 16 months	Median PFS: 43 months vs. 29 months	0.19
SWOG-1211 (Active,not recruiting)	II	Elo-VRD vs. VRD	100	Median PFS: 31.5 months vs. 33.6 months	-	-
MAIA (Completed)	III	Dara-RD vs. RD	92	Median PFS: 45 months vs. 29 months	Median PFS: 61.9 months vs. 34.9 months	NS
ALCYONE (Completed)	III	Dara-VMP vs. VMP	98	Median PFS: 18 months vs. 18 months	3-year PFS: 50.7% vs. 18.5% 3-year OS: 78.0% vs. 67.9%	NS
GMMG-CONCEPT (Active,not recruiting)	II	Isa-KRD	125	MRD^−^ rate in transplant eligible: 67.7% MRD^−^ rate in transplant ineligible: 54.2%		-
IMROZ (Active,not recruiting)	III	Isa-VRD vs. VRD	74	Hazard ratio: 0.97		NS
CEPHEUS (Active,not recruiting)	III	Dara-VRD vs. VRD	52	Hazard ratio: 0.88		NS

**Table 6 cells-14-00776-t006:** Outcomes in high genetic risk relapsed/refractory MM patients.

Study	Phase	Regimens	N. HRMM Patients	Outcomes in HRMM	General Outcomes	*p*-Value
ASPIRE (Completed)	III	KRD vs. RD	100	Median PFS: 23 months vs. 13.9 months	Median PFS: 29.6 months vs. 19.5 months	NS
APOLLO (Unknown status)	III	Dara-PD vs. PD	74	Median PFS: 5.8 months vs. 4 months	Median PFS: 12.4 months vs. 6.9 months	NS
POLLUX (Completed)	III	Dara-RD vs. RD	70	Median PFS: 26.8 months vs. 8.3 months	Median PFS: 44.5 months vs. 17.5 months	-
TOURMALINE-MM1 (Completed)	III	Ixa-RD vs. RD	137	Median PFS: 21.4 months vs. 9.7 months	Median PFS: 20.6 months vs. 14.7 months	<0.05
CANDOR (Completed)	III	Dara-KD vs. KD	74	Median PFS: 11.2 months vs. 7.4 months	Median PFS: 28.6 months vs. 15.2 months	NS
IKEMA (Completed)	III	Isa-KD vs. KD	73	Median PFS: not reached vs. 18.2 months	Median PFS: 35.7 months vs. 19.2 months	NS
DREAMM-7 (Active, not recruiting)	III	Belantamab mafodotin-VD vs. VD	136	Median PFS: 33.2 months vs. 10.5 months	Median PFS: 36.6 months vs. 13.4 months	−
BOSTON (Completed)	III	SVD vs. VD	256	ORR: 78% vs. 57% Median PFS: 12.9 months vs. 8.6 months	Median PFS: 13.9 months vs. 9.4 months	<0.05
ICARIA (Completed)	III	Isa-PD vs. PD	60	ORR: 50% vs. 16.7% Median PFS: 7.5 months vs. 3.7 months		NS

**Table 7 cells-14-00776-t007:** Outcomes with T-cell redirecting therapies in MM patients.

Study	Phase	Regimens	N. HRMM Patients	Outcomes in HRMM	General Outcomes	*p*-Value
MajesTEC-1 (Active, not recruiting)	I-II	Teclistamab	43	No benefits	ORR: 63% Median PFS: 11.3 months	−
MagnetisMM-3 (Active, not recruiting)	II	Elranatamab	31	ORR: 54.8% 12-month DOR: 57%	ORR: 61% Median PFS: 13.4 months	−
MonumenTAL-1 (Active, not recruiting)	I-II	Talquetamab	18	ORR: 55.6-66.7%	ORR: 64-70%	−
RedirecTT-1 (Active, not recruiting)	Ib	Teclistamab + talquetamab	15	-	ORR: 84% Median PFS: 20.9 months	−
KarMMa-1 (Active, not recruiting)	II	Ide-cel	45	Median PFS: 10.4 months	Median PFS: 8.2 months	−
CARTITUDE-1 (Completed)	Ib-II	Cilta-cel	13	ORR: 100% Median PFS: 21.1 months	ORR: 97.9% Median PFS: 34.9 months	−
CARTITUDE-4 (Active, not recruiting)	III	Cilta-cel vs. dara-PD/PVD	255	ORR: 88.5% Benefit in favor of cilta-cel	ORR: 84.6% vs. 67.3% 1-year PFS: 75.9% vs. 48.6%	<0.05

## Data Availability

No new data were created or analyzed in this study.

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
