# Peer review of "High-Risk Genetic Multiple Myeloma: From Molecular Classification to Innovative Treatment with Monoclonal Antibodies and T-Cell Redirecting Therapies"

_cells, 2025, doi:10.3390/cells14110776_

Round 1
Reviewer 1 Report
Comments and Suggestions for Authors
The manuscript entitled: “High-risk genetic multiple myeloma: from molecular classification to
innovative treatment with monoclonal antibodies and T-cell redirecting therapies (ID: cells-3640636)” by De Novellis et al. aims to provide an overview of molecular insights and a therapeutic update in MM patients with high-risk genetic situations.
Albeit the review is well written and of special interest, comments should be addressed to further improve the manuscript.
Comments:
- Page 3, section 3. Please provide a table of this updated model by Avet-Loiseau for a better overview.
- Page 7, section 6. Please provide also more information about elderly MM patients with HRMM in this section.
- Page 9, section 8: This section should be more enlarged due to novel approaches e.g. within therapy with anti-BCMA BiTES and CAR-T-cell therapy.
- Page 10, section conclusion: This section should be enlarged according to 1) what are next important future goals, 2) what is the unmet clinical need. Moreover, please provide more information how e.g. MRD-guided treatment algorithms or the new model be Avet-Loiseau could be routinely implemented.
- Page 7, section 6: line 228. Please provide additional references where appropriate.
Author Response
The manuscript entitled: “High-risk genetic multiple myeloma: from molecular classification to innovative treatment with monoclonal antibodies and T-cell redirecting therapies (ID: cells-3640636)” by De Novellis et al. aims to provide an overview of molecular insights and a therapeutic update in MM patients with high-risk genetic situations. Albeit the review is well written and of special interest, comments should be addressed to further improve the manuscript.
Response to General Comments. We thank the Reviewer for the positive feedback on our manuscript and for helpful comments that have markedly improved our work.
Comment 1. Page 3, section 3. Please provide a table of this updated model by Avet-Loiseau for a better overview.
Response to Comment 1. Please refer to the new Table 2, as shown below.
Table 2. Risk stratification systems.
Stage |
ISS |
R-ISS |
I |
Sβ2M < 3.5 mg/l Serum albumin ≥ 3.5 g/dl |
Sβ2M < 3.5 mg/l Serum albumin ≥ 3.5 g/dl Standard-risk CA by iFISH Normal LDH |
II |
Sβ2M > 3.5 mg/l and serum albumin < 3.5 g/dl OR 3.5 mg/l < Sβ2M > 5.5 mg/l |
Not R-ISS stage I or III |
III |
Sβ2M ≥ 5.5 mg/L |
Sβ2M ≥ 5.5 mg/L and either High-risk CA by FISH OR High LDH |
Genetic risk |
||
|
R-ISS Standard-risk CA |
Deletion of chromosome 17, or 17p-, translocation of chromosomes 14 and 16, or t(14;16), and translocation of chromosomes 4 and 14, or t(4;14) |
|
EMMA High genetic risk CA |
del(17p) > 20% TP53 mutation Biallelic del1p32 1q gain and monoallelic del1p32 t(4;14) or t(14;16) or t(14;20) and either 1q gain or monoallelic del1p32 |
Comment 2. Page 7, section 6. Please provide also more information about elderly MM patients with HRMM in this section.
Response to Comment 2. We thank the Reviewer for having pointed out this missing information.
On page 9, the following test was added “However, in the phase III IMROZ trial, isatuximab plus VRD (isa-VRD) has not shown improved outcomes in patients with high-risk cytogenetic features and older age [84], as well as daratumumab plus VRD in the CEPHEUS study [85].”
Table 5 (former Table 4) has been updated as follows.
Table 5. First-line treatments in ASCT-ineligible MM patients.
Study |
Phase |
Regimens |
N. HRMM patients |
Outcomes in HRMM |
General outcomes |
P value |
SWOG S0777 (Active, not recruiting) |
III |
VRD vs RD |
104 |
Median PFS: 38 months vs 16 months |
Median PFS: 43 months vs 29 months |
0.19 |
SWOG-1211 (Active, not recruiting) |
II |
Elo-VRD vs VRD |
100 |
Median PFS: 31.5 months vs. 33.6 months |
- |
- |
MAIA (Completed) |
III |
Dara-RD vs RD |
92 |
Median PFS: 45 months vs 29 months |
Median PFS: 61.9 months vs 34.9 months |
NS |
ALCYONE (Completed) |
III |
Dara-VMP vs VMP |
98 |
Median PFS: 18 months vs 18 months |
3-year PFS: 50.7% vs. 18.5% 3-year OS: 78.0% vs 67.9% |
NS |
GMMG-CONCEPT (Active, not recruiting) |
II |
Isa-KRD |
125 |
MRD- rate in transplant eligible: 67.7% MRD- rate in transplant ineligible: 54.2% |
- |
|
IMROZ (Active, not recruiting) |
III |
Isa-VRD vs VRD |
74 |
Hazard ratio: 0.97 |
NS |
|
CEPHEUS (Active, not recruiting) |
III |
Dara-VRD vs VRD |
52 |
Hazard ratio: 0.88 |
NS |
Comment 3. Page 9, section 8: This section should be more enlarged due to novel approaches e.g. within therapy with anti-BCMA BiTES and CAR-T-cell therapy.
Response to Comment 3. We thank the Reviewer for this point of discussion, and we have implemented this section as follows.
“Innovative T-cell redirecting therapies targeting surface antigens on malignant plasma cells include the family of (BiTEs) and CAR-T cells [98]. Currently, two anti-BCMA BiTEs, teclistamab and elranatamab, and one anti-GPRC5D BiTE, talquetamab, are approved for treatment of relapsed/refractory MM. Data on their efficacy in HRMM patients remains limited, as only a small proportion of them are in-cluded in clinical trials (Table 7) [98]. For example, HRMM patients treated with teclistamab or elranatamab show slightly worse outcomes compared to the overall population, as described in the phase I-II MajesTEC-1 trial and in the phase II Mag-netisMM-3 study, respectively [99], [100], while talquetamab induces similar ORR between high- and standard-risk genetic populations, as documented in the Monu-menTAL-1 study [101]. Combination of teclistamab or elranatamab with daratumumab could be more effective; however, updated data on HRMM subgroups are not available yet [102], [103], [104]. Of particular interest is the combination of teclistamab and talquetamab in the RedirecTT-1 study, enrolling 63 patients with triple-refractory MM. Although results for HRMM subgroup are not yet available, they are highly expected due to the innovative nature of this dual BiTE approach targeting distinct anti-gens [105]. Moreover, several trials are currently ongoing to assess the efficacy of BiTEs combined with other anti-MM agents, such as NCT05243797, NCT05083169, NCT05090566, NCT04649359, NCT05317416, NCT04798586, and NCT05228470.
CAR-T cells, autologous genetically engineered T lymphocytes, are modified to express a chimeric antigen receptor that recognizes BCMA antigen, mainly on neo-plastic plasma cells. In MM, two anti-BCMA CAR-T cell products are currently ap-proved: idecabtagene vicleucel (ide-cel) and ciltacabtagene autoleucel (cilta-cel) [98]. Ide-cel, the first FDA-approved anti-MM CAR-T product, has shown efficacy also in HRMM, although with lower ORR and shorter PFS compared to standard-risk patients, as shown in the phase 2 KarMMa study [106][74]. In HRMM patients with cilta-cel after 3 or more prior lines of therapy, ORR is also impressive; however, duration of response and PFS are shorter compared to standard risk, as described in the CARTI-TUDE-1 trial [107], [108]. Moreover, cilta-cel with PVd or DPd has comparable efficacy in high and standard genetic risk MM patients, as observed in the phase 3 CARTI-TUDE-4 study [109]. Several clinical trials are currently ongoing to evaluate the efficacy of CAR-T cell products in earlier lines of therapy, including in HRMM, such as in NCT04923893, NCT05257083, NCT05393804, and NCT06045806 trials.
Despite these advancements and the undeniable efficacy of T-cell redirecting therapies, several real-world barriers reduce their wide use in HRMM patients, also as earlier treatment lines. Indeed, toxicity management remains complex, as cytokine re-lease syndrome and neurotoxicity require specialized and experienced centers for quick identification and treatment. Additionally, access to CAR-T cells and BiTEs is limited by manufacturing logistics, center capacity, and costs, which currently negatively impact their early use in routine clinical practice [110]. Moreover, the lack of long-term follow-up data in HRMM does not permit yet to assess cost-efficacy of these therapies, thus adding evidence to support their early use in this high-risk population [98].”
Comment 4. Page 10, section conclusion: This section should be enlarged according to 1) what are next important future goals, 2) what is the unmet clinical need. Moreover, please provide more information how e.g. MRD-guided treatment algorithms or the new model be Avet-Loiseau could be routinely implemented.
Response to Comment 4. We thank the Reviewer for this point, and we have modified the Conclusions section as follows.
“In recent years, significant advancements have been made in MM treatment, with the introduction of several novel therapeutic agents for newly diagnosed patients, regardless of ASCT eligibility, as well as for those with relapsed/refractory disease. At the same time, increasing emphasis has been placed for a more precise risk stratification system, by incorporating not only clinical and laboratory biomarkers, but also molecular and cytogenetic alterations [20]. Indeed, HRMM has emerged as a distinct biological and clinical entity, characterized by aggressive behavior and poor prognosis, often driven by complex genomic and transcriptomic signatures. Despite important therapeutic and diagnostic innovations, prognosis for these subjects remains poor, likely due to different pathogenetic mechanisms underlying this type of disease. For this reason, a paradigm shift is essential for improving clinical management of HRMM, as future strategies should focus on refining risk stratification models and to tailor therapies based on specific patient’s molecular features. For example, the risk-adapted model proposed by Avet-Loiseau offers a more accurate classification of HRMM and should be prospectively validated for clinical implementation. Indeed, this model could enable early identification of ultra–high-risk patients and help tailor treatment intensity accordingly. In addition, prognostic impact of specific lesions, such as biallelic TP53 in-activation or 1p32 deletions, must be clearly delineated in clinical trials to guide treatment escalation. Therefore, it is essential to improve current risk stratification systems, not only including cytogenetic abnormalities and only TP53 mutational status, while also considering clonal hematopoiesis and other pathogenetic variants, as also described in myeloid and lymphoid malignancies. Moreover, routine integration of MRD detection into treatment algorithms represents a critical future direction, as MRD negativity is a strong predictor of long-term outcomes, particularly in HRMM. In this subset of MM patients, durable responses are less common, and MRD could serve as a surrogate to modulate therapy, allowing intensification for non-responders or de-escalation to reduce toxicity in deep responders. However, prospective trials are urgently needed to validate MRD-guided treatment decisions, especially in high-risk settings.
While intensified therapeutic approaches are now accepted standards, they alone are not sufficient. Integration of immunotherapy, particularly T-cell redirecting agents, such as BiTEs and CAR-T cells, earlier into frontline treatment, in a genetically risk-adapted and guided approach, would be a desirable strategy to overcome traditional resistance mechanisms. These agents, ideally combined with checkpoint inhibitors or targeted molecules, could provide synergistic effects. However, barriers related to toxicity, accessibility, cost, and logistical complexity remain substantial, particularly when considering frontline or early relapse application. Ultimately, addressing the needs of HRMM will require a multi-omics-driven approach coupled with prospective clinical trials specifically designed for this subgroup, rather than relying on post hoc analyses from broader studies. Only through this tailored strategy we can significantly modify the natural history of HRMM and offer a real opportunity for long-term dis-ease control in this group of patients.”
Comment 5. Page 7, section 6: line 228. Please provide additional references where appropriate.
Response to Comment 5. We have added the following reference: Sonneveld P, Dimopoulos MA, Boccadoro M, Quach H, Ho PJ, Beksac M, Hulin C, Antonioli E, Leleu X, Mangiacavalli S, Perrot A, Cavo M, Belotti A, Broijl A, Gay F, Mina R, van de Donk NWCJ, Katodritou E, Schjesvold F, Balari AS, Rosiñol L, Delforge M, Roeloffzen W, Silzle T, Vangsted A, Einsele H, Spencer A, Hajek R, Jurczyszyn A, Lonergan S, Ahmadi T, Liu Y, Wang J, Vieyra D, van Brummelen EMJ, Vanquickelberghe V, Sitthi-Amorn A, de Boer CJ, Carson R, Rodriguez-Otero P, Bladé J, Moreau P. A plain language summary of the PERSEUS study of daratumumab plus bortezomib, lenalidomide, and dexamethasone for treating newly diagnosed multiple myeloma. Future Oncol. 2024;20(38):3043-3063. doi: 10.1080/14796694.2024.2394323. Epub 2024 Sep 17. PMID: 39287147.
Reviewer 2 Report
Comments and Suggestions for Authors
Refer attachment.

Author Response
Overview
Review article discusses HRMM, emphasizing the poor prognosis associated with specific cytogenetic abnormalities like del(17p), TP53 mutations, and biallelic 1p32 deletions, despite advancements in therapy. It also highlights recent updates in risk stratification and evolving treatments, particularly monoclonal antibodies and T-cell redirecting therapies, aiming to improve outcomes through MRD-driven and genetically tailored strategies. Authors conclude that integrating early immunotherapy, refined molecular classifications, and next-generation T-cell therapies is essential to overcoming the challenges posed by HRMM.
Major Comments
Comment 1. Abstract is comprehensive but overly dense. It lacks conciseness and does not clearly highlight key findings or recommendations. Authors are requested to condense sentences and clearly delineate background, objectives, findings, and conclusions. Further, explicitly state the clinical or research implications of the review.
Response to Comment 1. We thank the Reviewer for this helpful comment, and we have reorganized the abstract as follows.
“High-risk genetic multiple myeloma (HRMM) remains a major therapeutic challenge, as patients harboring adverse genetic abnormalities, such as del(17p), TP53 mutations, and biallelic del(1p32), continue to experience poor outcomes despite recent therapeutic advancements. This review explores the evolving definition and molecular features of HRMM, focusing on recent updates in risk stratification and treatment strategies. New genetic classification proposed at the 2025 EMMA meeting offers improved prognostic accuracy, and supports more effective, risk-adapted treatment planning. In transplant-eligible patients, intensified induction regimens, tandem autologous stem cell transplantation, and dual-agent maintenance have shown improved outcomes, particularly when sustained minimal residual disease negativity is achieved. Conversely, in the relapsed or refractory setting, novel agents have demonstrated encouraging activity, although their specific efficacy in HRMM is under investigation. Moreover, treatment paradigms are shifting toward earlier integration of immunotherapy, and therapeutic strategies are individualized based on refined molecular risk profiles and clone dynamics. Therefore, a correct definition of HRMM could help in significantly improving both clinical and therapeutic management of a subgroup of patients with an extremely aggressive disease.”
Comment 2. Introduction: Good background information, but lacks a definitive problem statement and rationale for the review. Please clarify the knowledge gap this review addresses compared to existing literature. Also, please define HRMM and FHRMM more distinctly and early on.
Response to Comment 2. We thank the Reviewer for this clarification.
On pages 1-2, the following text was added “These patients typically exhibit aggressive disease biology driven by specific high-risk genetic abnormalities, and they represent a clinically distinct subset, defined as high-risk multiple myeloma (HRMM), that could overlap with a recently introduced entity, the functional high-risk multiple myeloma (FHRMM), which includes patients with early relapse or suboptimal response to therapies. [3]. To date, the presence of these adverse prognostic high-risk genetic features remains a major unmet clinical challenge, even in the era of novel agents. While several reviews have discussed therapeutic options in MM, few have comprehensively addressed the specific challenges posed by HRMM, such as risk classification systems and specific therapeutic strategies. Therefore, in this review, we aim at. (i) outlining the historical evolution and updated definitions of HRMM; (ii) at summarizing current therapeutic landscapes, including the role of monoclonal antibodies and T-cell redirecting therapies; and (iii) at highlighting future directions, such as molecularly-guided MRD-adapted treatment approaches.”
Comment 3. Section 2: Table 1 is helpful but lacks clarity regarding the prognostic stratification across combinations of abnormalities. It also relies heavily on frequency data without integrating clinical impact or therapeutic implications. Please include a brief commentary on the clinical management relevance of each abnormality. Furthermore, clarify distinctions between monoallelic vs biallelic impact, especially for del(1p32) and del(17p).
Response to Comment 3. We thank the Reviewer for this point of discussion. Please refer to the new Table 1 where we have added clinical and therapeutic implications and emphasized the difference between monoallelic and biallelic abnormalities for del1p32 and del 17p.
Table 1. High-risk genetic in multiple myeloma.
Abnormality |
Frequency |
Gene/Pathway |
Prognostic significance |
Clinical impact |
All 14q32 (IGH) t(4;14)
t(14;16)
t(14;20) |
45-50% 10% to 15%
<5%
<5% |
FGFR3/MMSET Upregulation MAF overexpression
MAFB overexpression |
Poor
Uncertain; mainly poor
Uncertain; mainly poor |
Rapid progression. Double ASCT
Double ASCT especially when associated with HR abnormalities Double ASCT especially when associated with HR abnormalities |
1q21 gain 2-3 copies ≥4 copies |
40% 20-30% 5-20% |
CKS1B, MCL1, ADAR1 overexpression |
Intermediate Poor |
Aggressive with organ failure Double ASCT especially when associated with HR abnormalities |
1p32 deletion Monoallelic
Biallelic |
10% |
FAF1/CDKN2C deficit |
Poor
Highly poor |
Double ASCT especially when associated with HR abnormalities Double ASCT + intensive maintenance |
del(17p)/TP53 mutation Single hit
Double hit |
8-15%
Deletion
Deletion + mutation |
TP53 |
Poor
Highly poor |
Poor sensitivity to therapy
Double ASCT + intensive maintenance |
Comment 4. Section 3: Valuable discussion but highly technical and reliant on EMMA meeting results with minimal references. Please include a comparative figure or summary table showing ISS vs R-ISS vs new classification. Also, clarify the practical implications of adopting the new risk model in clinical settings.
Response to Comment 4. Please refer to the new Table 2. We have also added clinical implications of the new risk stratification system. Additionally, we added the following sentence: “If definitively validated, the new genetic risk model will allow a deeper understanding of HRMM and more intensive therapies to be proposed to specific patient subgroups in the future.”
Table 2. Risk stratification systems.
Stage |
ISS |
R-ISS |
I |
Sβ2M < 3.5 mg/l Serum albumin ≥ 3.5 g/dl |
Sβ2M < 3.5 mg/l Serum albumin ≥ 3.5 g/dl Standard-risk CA by iFISH Normal LDH |
II |
Sβ2M > 3.5 mg/l and serum albumin < 3.5 g/dl OR 3.5 mg/l < Sβ2M > 5.5 mg/l |
Not R-ISS stage I or III |
III |
Sβ2M ≥ 5.5 mg/L |
Sβ2M ≥ 5.5 mg/L and either High-risk CA by FISH OR High LDH |
Genetic risk |
||
|
R-ISS Standard-risk CA |
Deletion of chromosome 17, or 17p-, translocation of chromosomes 14 and 16, or t(14;16), and translocation of chromosomes 4 and 14, or t(4;14) |
|
EMMA High genetic risk CA |
del(17p) > 20% TP53 mutation Biallelic del1p32 1q gain and monoallelic del1p32 t(4;14) or t(14;16) or t(14;20) and either 1q gain or monoallelic del1p32 |
Comment 5. Section 4: Excellent depth, but the content or information is dense with overlapping mechanisms. It lacks clear linkage between molecular aberrations and therapeutic resistance.
Response to Comment 5. We thank the Reviewer for the positive feedback on this section, and we have shortened the first part and added links between molecular changes and pathological mechanisms, as shown below.
“In HRMM, the presence of specific molecular alterations confers aggressive disease behavior, drug resistance, and clonal evolution [19], that can be pharmacologically targeted. Del(17p), with or without TP53 mutations, results in the loss of p53 tumor suppressor function, promoting genomic instability, impaired DNA repair, and apoptosis resistance, thereby contributing to unresponsiveness to conventional therapies [20]. Gain or amplification of chromosome 1q21, another hallmark of HRMM, involves several oncogenes, which accelerates cell cycle progression, drug resistance, and adverse outcomes [21]. IGH translocations lead to altered chromatin structure and epigenetic regulation, DNA damage tolerance, resistance to proteasome inhibitors, cell adhesion, migration, and angiogenesis (Figure 1) [22][12]. These molecular events not only drive disease progression, but also induce an immunosuppressive microenvironment, impacting the efficacy of monoclonal antibodies and T-cell redirecting agents. A deeper understanding of these mechanisms is essential for the development of targeted therapeutic strategies tailored to genetic and molecular landscape of HRMM [23], [24].
In detail, there are different pathogenetic events occurring at disease initiation and at progression (Figure 2) [25][26].
4.1 IGH translocations
Translocations involving the IGH loci pose place various oncogenes next to the strong enhancer region of these immunoglobulin regiones, that are highly active in mature B cells, thus translocated oncogenes result in increased expression, ultimately resulting in cell cycle dysregulation and proliferation, and reduced DNA repair ability [PMID: 24327604; PMID: 31323969]. IGH translocations and hyperdiploidy usually are present at disease initiation and lead to cell cycle dysregulation, by affecting CDK4 and CDK6 complexes that favor the dissociation transcription factor E2F by RB phosphorylation [27], [28], [29]. Consequently, E2F concurs to transcription of genes involved in G1/S checkpoint step, and to upregulation of FGFR3 and MMSET (also known as NSD2) [30], [31]. This latter, a histone methyltransferase, highly influences methylation status of cells by increasing H3K36me2 and reducing H3K27me3 [32]. In neo-plastic plasma cells, MMSET upregulation leads to altered methylation, cell adhesion, increased proliferation and survival, and genomic instability [33], [34][35]. Moreover, MMSET can promote non-homologous end-joining at deprotected telomeres, altering DNA repair process [PMID: 35534749]. Other IGH translocations, including t(14;16) and t(14;20), are associated with overexpression of the oncogenes MAF and MAFB, integrins such as integrin β7, and apolipoprotein B mRNA editing enzyme catalytic subunit-induced mutation signature [36], [37]. All these alterations are linked to tumor invasion, metastasis, higher ability to resist starvation and stress, and increased ge-nomic and chromosomal instability [PMID: 35534749].
4.2 Methylation
HDAC6 inhibition results in high pro-apoptotic effects, likely because of a concomitant modulation of protein degradation. HDAC6 binds to polyubiquitinated proteins and facilitates the removal of protein aggregates by regulating aggresome formation and their autophagic degradation through HSF1 and HSP90 activation [35]. This abnormal methylation status in neoplastic plasma cells is also caused by increased expression of the histone methyltransferase EZH2 [28]. Moreover, mutations in DNA methylation modifiers, such as in IDH1, can also occur, especially during disease progression, and can contribute to global gene expression alteration [28].
4.3 MYC translocations
Secondary events at disease progression are MYC translocations, likely promoted by genes with super-enhancers active in late B cell stages [38]. Copy number changes usually occur at 8q24 and are associated with MYC translocation in 30% of cases [39], [40].
4.4 Chromosome 1 abnormalities
Amplification of 1q is also common, and starts from the formation of dicentric chromosomes, leading to multiple breakage–fusion–bridge cycles with consequent gene at 1q21[41]. This chromosomal region comprises of several genes, including CKS1B, MCL1 encoding for a BCL‑2 family member, ANP32E, and ILF2 encoding for a protein required for RNA splicing of genes for DNA damage repair proteins [42]. These alterations are probably induced by hypoxia and aberrant expression of KDM4A, as also demonstrated by a significant association between RAS mutations, loss of p53 function, and upregulation of HIF1α and lactate dehydrogenase A [43], [44]. Conversely, loss of 1p leads to deletion of CDKN2C, FAF1, FAM46C, RPL5, and ecotropic viral integration site 5 [45]. In particular, CDKN2A, together with CDKN2B, CDKN2C and CDKN2D, regulate cell proliferation by inhibiting the activity of cyclin D–CDK complexes [46]. Their loss of function can derive from 1p deletion, homozygous inactivation of RB1, and/or DNA methylation-mediated silencing, and results in increased cell proliferation alongside with RAS pathway upregulation [28]. Therefore, chromo-some 1 abnormalities result in cell cycle dysregulation, proliferation, anti-apoptotic signaling activation, and cell growth [PMID: 35534749].”
Comment 6. Please include subheadings or bullet points to delineate individual pathways (e.g., TP53 loss, 1q21 gain).
Response to Comment 6. Subheadings for section 4 have been added as suggested.
Comment 7. Section 5 and 6: Numerous trials are included, but the narrative is scattered and lacks constructive framework. Also, there is minimal discussion on the limitations of current evidence for HRMM subgroups. Authors can provide summary tables comparing regimens with HRMM-specific outcomes. Also, can include discussion on limitations in trial design, such as underrepresentation of HRMM patients.
Response to Comment 7. We thank the Reviewer for pointing out missing discussion, that we have added on each section, as described below. For summary tables, we would like to keep our table format, where we have compared total cohorts results to HRMM group for each trial and based on category (e.g., transplant eligible, and so on).
On page 8, the section 5 has been reorganized as follows.
“Consolidation therapy is defined as 2-4 additional cycles of treatment using the same agents employed during induction followed by ASCT, and its role in MM management remains debated. For example, VRD-based regimens have been proposed as consolidation, because some studies, like EMN02/HO95, display improved outcomes, while others, such as the StaMINA trial, have not confirmed this benefit [60], [62], [64]. However, current clinical guidelines recommend the use of consolidation therapy in selected cases, such as patients with persistent MRD positivity after ASCT or those with HRMM. Given the limited and heterogeneous evidence regarding consolidation, maintenance therapy has emerged as a critical component in post-transplant management, particularly in HRMM, where sustained disease control is essential for im-proving long-term outcomes. Maintenance treatment is less intensive than induction, typically administered orally until MM progression or intolerance, with the primary aim of delaying relapse [65], [66]. Lenalidomide alone has shown clear benefits in PFS and OS in the general MM population [67], although its efficacy in HRMM appears limited in prolonging PFS. Therefore, lenalidomide in combination with bortezomib, with or without dexamethasone, could offer added benefit in this subgroup of patients [7], [68], [69]. In the CASSIOPEIA trial, single agent daratumumab has been shown to prolong PFS; however, no specific subgroup analysis was reported for HRMM [70]. Conversely, the randomized phase III TOURMALINE-MM3 study demonstrated that ixazomib as maintenance after ASCT could improve outcomes in HRMM group [71]. In the ongoing PERSEUS study (NCT03710603), maintenance with lenalidomide and daratumumab is being tested in both standard and high-genetic risk patients [54], [72]. Despite these promising results, clinical trials lack patients’ stratification by high-risk genetic features or only include limited analyses on HRMM disease, partly due to the underrepresentation of high-risk patients in trial populations. Additionally, the absence of MRD-guided treatment adaptation in several older pivotal studies limits their relevance in the current precision medicine landscape.”
On page 9, the section 6 has been updated as follows.
“In ASCT-ineligible patients, the therapeutic goal shifts away from achieving deep and sustained response, to balance treatment efficacy with tolerability and to preserve quality of life. However, evidence-guided treatment decisions in this setting are limited, particularly for frail patients with high-risk genetic abnormalities, who are often underrepresented in clinical trials (Table 5). Bortezomib in combination with IMiDs has been associated with hematological responses in HRMM patients [73]. In the phase III VISTA trial, VMP regimen demonstrated comparable survival rates to those observed in standard risk patients [74]. In contrast, continuous or fixed-duration RD therapy is ineffective in these patients [75]. However, adding bortezomib to RD as a VRD triplet can improve PFS [76], whereas the addition of elotuzumab did not show the same benefit [77]. Moreover, integration of anti-CD38 monoclonal antibodies into frontline regimens has yielded mixed results in HRMM. Indeed, dara-RD has documented superiority over RD in HRMM, although the adverse prognostic impact of high-risk genetics is not fully mitigated, as PFS remains shorter than that observed in the overall study population [78], [79], [80]. Similarly, the addition of daratumumab to VMP regimen did not improve outcomes compared to VMP in HRMM patients [81]. Interpretation of these results is limited by the small proportion of high-risk patients in the intention-to-treat populations, reducing the statistical power of subgroup analyses. To address this limitation, a meta-analysis of phase III studies has been conducted, showing that the addition of daratumumab could prolong PFS in HRMM patients [82].
Alternative proteasome inhibitor–based combinations have also been explored in this setting. Carfilzomib plus cyclophosphamide-dexamethasone (KCD) may mitigate genetical-related poor prognosis [83], and the quadruplet isa-KRD has been associated with encouraging MRD negativity rates at the end of consolidation [57]. However, in the phase III IMROZ trial, isatuximab plus VRD (isa-VRD) has not shown improved outcomes in patients with high-risk cytogenetic features and older age [84], as well as daratumumab plus VRD in the CEPHEUS study [85].”
Comment 8. Section 7: Good data compilation, however lacks critical appraisal of efficacy in HRMM vs standard-risk patients. Authors can include forest plots or comparative efficacy discussion for each class of drugs. Can also highlight unmet needs and propose ideal sequencing strategies for HRMM.
Response to Comment 8. We thank the Reviewer for these helpful comments, and we have added comparative efficacy discussion, as shown below.
“The prevalence of high-risk genetic features increases at MM progression, due to clonal evolution after prior therapies [86]. While triplet combinations generally out-perform doublets in both standard- and high-risk patients, their efficacy in HRMM remains suboptimal. Most regimens do not fully mitigate the adverse prognostic impact of these genetic abnormalities, and PFS consistently remains shorter compared to standard-risk patients (Table 6) [47]. According to ESMO guidelines, treatment selection is primarily based on lenalidomide sensitivity or refractoriness [47]. In lenalidomide-sensitive patients, triplet regimens, such as dara-RD, KRD, elotuzumab-RD, or ixazomib-RD, have shown clinical efficacy [87], [88], [89], [90], [91], although HRMM patients display inferior PFS and response durability compared to standard-risk cohorts. In lenalidomide refractory HRMM patients, preferred regimens are combinations of anti-CD38 monoclonal antibodies with PD (dara-PD) or KD (isa-KD) [92], [93], [94], [95]. In both phase III CASTOR and ICARIA trials, dara-VD and isa-PD have been associated with longer PFS in HRMM patients compared to control arms, although the strength of this benefit is lower than that observed in standard risk populations [24]. Emerging agents may provide incremental benefits in HRMM. The anti-BCMA anti-body-drug conjugate belantamab-mafodotin in combination with VD could be even superior to dara-VD in this subgroup of patients [96]. Similarly, the nuclear export inhibitor, selinexor, when combined with VD (SVD) has also demonstrated superiority over VD in HRMM, as shown in the phase III BOSTON trial [97].
Despite some clinical efficacy, these regimens are not curative, as current treatment options fail to achieve deep and durable responses comparable to those observed in standard-risk MM. Therefore, unmet needs persist across the treatment continuum for HRMM patients, and a rational treatment sequencing strategy would ideally prioritize the use of the most effective agents as earlier therapy lines, before clonal evolution would reduce drug responsiveness. Tailored approaches based on molecular profiles, coupled with MRD-adapted therapy escalation or de-escalation, could further enhance outcomes in this high-risk population.”
Comment 9. Section 8: Does not discusses real-world barriers (toxicity, access, durability of response). Please add data on response durability in HRMM vs standard-risk. Can additionally, discuss logistical and economic limitations in using CAR-T and BiTEs earlier in therapy.
Response to Comment 9. Please see below the updated section 8.
“Innovative T-cell redirecting therapies targeting surface antigens on malignant plasma cells include the family of (BiTEs) and CAR-T cells [98]. Currently, two anti-BCMA BiTEs, teclistamab and elranatamab, and one anti-GPRC5D BiTE, talquetamab, are approved for treatment of relapsed/refractory MM. Data on their efficacy in HRMM patients remains limited, as only a small proportion of them are included in clinical trials (Table 7) [98]. For example, HRMM patients treated with teclistamab or elranatamab show slightly worse outcomes compared to the overall population, as described in the phase I-II MajesTEC-1 trial and in the phase II Mag-netisMM-3 study, respectively [99], [100], while talquetamab induces similar ORR between high- and standard-risk genetic populations, as documented in the Monu-menTAL-1 study [101]. Combination of teclistamab or elranatamab with daratumumab could be more effective; however, updated data on HRMM subgroups are not available yet [102], [103], [104]. Of particular interest is the combination of teclistamab and talquetamab in the RedirecTT-1 study, enrolling 63 patients with triple-refractory MM. Although results for HRMM subgroup are not yet available, they are highly expected due to the innovative nature of this dual BiTE approach targeting distinct anti-gens [105]. Moreover, several trials are currently ongoing to assess the efficacy of BiTEs combined with other anti-MM agents, such as NCT05243797, NCT05083169, NCT05090566, NCT04649359, NCT05317416, NCT04798586, and NCT05228470.
CAR-T cells, autologous genetically engineered T lymphocytes, are modified to express a chimeric antigen receptor that recognizes BCMA antigen, mainly on neo-plastic plasma cells. In MM, two anti-BCMA CAR-T cell products are currently ap-proved: idecabtagene vicleucel (ide-cel) and ciltacabtagene autoleucel (cilta-cel) [98]. Ide-cel, the first FDA-approved anti-MM CAR-T product, has shown efficacy also in HRMM, although with lower ORR and shorter PFS compared to standard-risk patients, as shown in the phase 2 KarMMa study [106][74]. In HRMM patients with cilta-cel after 3 or more prior lines of therapy, ORR is also impressive; however, duration of response and PFS are shorter compared to standard risk, as described in the CARTI-TUDE-1 trial [107], [108]. Moreover, cilta-cel with PVd or DPd has comparable efficacy in high and standard genetic risk MM patients, as observed in the phase 3 CARTI-TUDE-4 study [109]. Several clinical trials are currently ongoing to evaluate the efficacy of CAR-T cell products in earlier lines of therapy, including in HRMM, such as in NCT04923893, NCT05257083, NCT05393804, and NCT06045806 trials.
Despite these advancements and the undeniable efficacy of T-cell redirecting therapies, several real-world barriers reduce their wide use in HRMM patients, also as earlier treatment lines. Indeed, toxicity management remains complex, as cytokine re-lease syndrome and neurotoxicity require specialized and experienced centers for quick identification and treatment. Additionally, access to CAR-T cells and BiTEs is limited by manufacturing logistics, center capacity, and costs, which currently negatively impact their early use in routine clinical practice [110]. Moreover, the lack of long-term follow-up data in HRMM does not permit yet to assess cost-efficacy of these therapies, thus adding evidence to support their early use in this high-risk population [98].”
Comment 10. The conclusion is well-written but reiterative. Conclusion should also emphasize future directions and research needs (e.g., multi-omics, prospective trials targeting HRMM).
Response to Comment 10. We thank the Reviewer for this helpful comment, and we have reorganized the conclusion as follows.
“In recent years, significant advancements have been made in MM treatment, with the introduction of several novel therapeutic agents for newly diagnosed patients, regardless of ASCT eligibility, as well as for those with relapsed/refractory disease. At the same time, increasing emphasis has been placed for a more precise risk stratification system, by incorporating not only clinical and laboratory biomarkers, but also molecular and cytogenetic alterations [20]. Indeed, HRMM has emerged as a distinct biological and clinical entity, characterized by aggressive behavior and poor prognosis, often driven by complex genomic and transcriptomic signatures. Despite important therapeutic and diagnostic innovations, prognosis for these subjects remains poor, likely due to different pathogenetic mechanisms underlying this type of disease. For this reason, a paradigm shift is essential for improving clinical management of HRMM, as future strategies should focus on refining risk stratification models and to tailor therapies based on specific patient’s molecular features. For example, the risk-adapted model proposed by Avet-Loiseau offers a more accurate classification of HRMM and should be prospectively validated for clinical implementation. Indeed, this model could enable early identification of ultra–high-risk patients and help tailor treatment intensity accordingly. In addition, prognostic impact of specific lesions, such as biallelic TP53 in-activation or 1p32 deletions, must be clearly delineated in clinical trials to guide treatment escalation. Therefore, it is essential to improve current risk stratification systems, not only including cytogenetic abnormalities and only TP53 mutational status, while also considering clonal hematopoiesis and other pathogenetic variants, as also described in myeloid and lymphoid malignancies. Moreover, routine integration of MRD detection into treatment algorithms represents a critical future direction, as MRD negativity is a strong predictor of long-term outcomes, particularly in HRMM. In this subset of MM patients, durable responses are less common, and MRD could serve as a surrogate to modulate therapy, allowing intensification for non-responders or de-escalation to reduce toxicity in deep responders. However, prospective trials are urgently needed to validate MRD-guided treatment decisions, especially in high-risk settings.
While intensified therapeutic approaches are now accepted standards, they alone are not sufficient. Integration of immunotherapy, particularly T-cell redirecting agents, such as BiTEs and CAR-T cells, earlier into frontline treatment, in a genetically risk-adapted and guided approach, would be a desirable strategy to overcome traditional resistance mechanisms. These agents, ideally combined with checkpoint inhibitors or targeted molecules, could provide synergistic effects. However, barriers related to toxicity, accessibility, cost, and logistical complexity remain substantial, particularly when considering frontline or early relapse application. Ultimately, addressing the needs of HRMM will require a multi-omics-driven approach coupled with prospective clinical trials specifically designed for this subgroup, rather than relying on post hoc analyses from broader studies. Only through this tailored strategy we can significantly modify the natural history of HRMM and offer a real opportunity for long-term dis-ease control in this group of patients.”
Minor Comments
Comment 11. Introduction: “To date” and “in the era of novel agents” are vague and could benefit from specific years or agents.
Response to Comment 11. Removed and rephrased where appropriate.
Comment 12. Figure 2 lacks descriptive legend. Please consider expanding caption for readability and clarity.
Response to Comment 12. Figure 2 legend has been expanded as follows “Figure 2. Pathways altered in MM. Principal pathways involved in high-risk disease, including NF‑κB, MAPK, cell cycle, hypoxia, and DNA-damage repair pathways. In details, the absence of TP53 results in the lack of cell cycle control through E2F1 and Cyclin D3, and cells continue to G1/S phase even though they carry DNA breaks, thus promoting genomic instability, impaired DNA repair, and apoptosis resistance. The t(4;14), t(14;16) and t(14;20) translocations lead to MMSET and MAF/MAFB overexpression, ultimately leading to increased E2F1 activation and translocation to the nucleus, where it promotes the transcription of genes involved in cell proliferation. Similarly, hyperdyploidy and t(11;14) promote cell cycle progression and proliferation through Cyclin D1/CDK4/9 modulation. Upregulation and/or activating mutation in FGFR3, KRAS, NRAS, and/or BRAF, and MYC translocations induce cell proliferation by direct gene transcription or by indirectly modulating cyclin activities, as well as TRAF3 and NF-κB. Moreover, altered BCR signaling is related to impaired apoptosis through modulation of IRF4, BCL-6, and PAX5. Finally, hypoxia can also influence RNA splicing of genes for DNA damage repair proteins, via upregulation of HIF1α and lactate dehydrogenase A. Figure made using https://smart.servier.com/.”
Comment 13. In tables, indicate whether outcome differences are statistically significant.
Response to Comment 13. We thank the Reviewer for this point, and we have added a column “P value” to show statistical significance for Tables 3-7.
We thank the Reviewer for the suggestion. When available, we implemented the statistical significance figure by adding a column to the tables
Comment 14. When listing ongoing trials, it is unclear if they are actively recruiting or completed—add status if known.
Response to Comment 14. We thank the Reviewer for this point. We added this information for each study in the tables.
Remark
Overall, the review offers significant insights. However, there is considerable room for significant improvement.
Response to Remark. We thank the Reviewer for insightful and helpful comments, that have markedly improved the quality of our work. We hope we have addressed all concerns.